# MitoTrace: A Computational Framework for Analyzing Mitochondrial Variation in Single-Cell RNA Sequencing Data

**DOI:** 10.3390/genes14061222

**Published:** 2023-06-04

**Authors:** Mingqiang Wang, Wankun Deng, David C. Samuels, Zhongming Zhao, Lukas M. Simon

**Affiliations:** 1Center for Precision Health, School of Biomedical Informatics, The University of Texas Health Science Center at Houston, Houston, TX 77030, USA; timwang5@stanford.edu (M.W.); wankun.deng@uth.tmc.edu (W.D.); 2Stanford Cardiovascular Institute, Stanford University School of Medicine, Stanford, CA 94305, USA; 3Vanderbilt Genetics Institute, Vanderbilt University School of Medicine, Nashville, TN 37232, USA; david.c.samuels@vanderbilt.edu; 4Department of Molecular Physiology & Biophysics, Vanderbilt University School of Medicine, Nashville, TN 37232, USA; 5Human Genetics Center, School of Public Health, The University of Texas Health Science Center at Houston, Houston, TX 77030, USA; 6MD Anderson Cancer Center UTHealth Graduate School of Biomedical Sciences, Houston, TX 77030, USA; 7Department of Biomedical Informatics, Vanderbilt University Medical Center, Nashville, TN 37203, USA; 8Therapeutic Innovation Center, Baylor College of Medicine, Houston, TX 77030, USA; 9Department of Biochemistry and Molecular Biology, Baylor College of Medicine, Houston, TX 77030, USA

**Keywords:** mitochondrial genetic variation, heteroplasmy, single-cell RNA sequencing, lineage tracing, R package

## Abstract

Genetic variation in the mitochondrial genome is linked to important biological functions and various human diseases. Recent progress in single-cell genomics has established single-cell RNA sequencing (scRNAseq) as a popular and powerful technique to profile transcriptomics at the cellular level. While most studies focus on deciphering gene expression, polymorphisms including mitochondrial variants can also be readily inferred from scRNAseq. However, limited attention has been paid to investigate the single-cell landscape of mitochondrial variants, despite the rapid accumulation of scRNAseq data in the community. In addition, a diploid context is assumed for most variant calling tools, which is not appropriate for mitochondrial heteroplasmies. Here, we introduce MitoTrace, an R package for the analysis of mitochondrial genetic variation in bulk and scRNAseq data. We applied MitoTrace to several publicly accessible data sets and demonstrated its ability to robustly recover genetic variants from scRNAseq data. We also validated the applicability of MitoTrace to scRNAseq data from diverse platforms. Overall, MitoTrace is a powerful and user-friendly tool to investigate mitochondrial variants from scRNAseq data.

## 1. Introduction

RNA sequencing (RNAseq) is one of the most popular molecular profiling modalities. Its versatility enables diverse analyses including differential gene expression, alternative splicing, allele-specific expression, alternative polyadenylation, copy number variation and variant calling, among others [1].

Previous studies have shown that bulk RNAseq data can be used to infer mitochondrial DNA (mtDNA) alterations and mtDNA mutations have been linked to a plethora of human diseases [2,3]. Mitochondrial diseases, in fact, represent the most prevalent category of inherited metabolic disorders and rank among the frequently encountered types of inherited neurological disorders [4]. For example, MELAS syndrome (mitochondrial encephalopathy, lactic acidosis, and stroke-like episodes) is caused by a mutation in the mitochondrial gene MT-TL1, which is responsible for encoding a transfer RNA molecule essential for protein synthesis within mitochondria. This mutation leads to impaired energy production and affects various organs, particularly the brain, resulting in symptoms such as recurrent strokes, seizures, muscle weakness, and cognitive impairments [5].

Therefore, computational frameworks for the study of mtDNA in RNAseq data will significantly expand analytical possibilities and may lead to novel insights which improve our understanding of biology and human disease. 

Since mitochondria carry their own mtDNA and hundreds to thousands of mtDNA copies exist in each individual cell [6], different mitochondrial polymorphisms can co-exist within an individual cell. These polymorphisms are referred to as mitochondrial heteroplasmies. The level of these heteroplasmies plays a crucial role in determining the extent to which mitochondrial mutations are pathological [7,8]. For example, the MELAS disease burden is strongly associated with the heteroplasmy level for 3243A>G in blood [9].

Previous work has shown that the mitochondrial heteroplasmy detection limit in bulk RNAseq is about 0.1% [10]. Due to the increased mutation rate and high RNAseq read coverage of the mitochondrial genome, recent studies have leveraged mitochondrial heteroplasmies for in silico lineage tracing of single cells. 

For example, Xu et al. developed EMBLEM for cell lineage tracing using mitochondrial mutations derived from ATAC-seq data, and showed that cell lineage can be reconstructed using the mitochondrial mutation profiles at single-cell resolution [11]. Similarly, Leif et al. demonstrated the use of mitochondrial mutations in single cells for in silico lineage tracing [12]. Later, Lareau et al. developed mitochondrial single-cell ATAC-seq for mtDNA genotyping and chromatin profiling, and implemented mGATK, a computational method for extracting mitochondrial genetic variation [13]. The authors applied their method to clonally trace thousands of cancer cells to establish connections between epigenomic variability and subclonal evolution inferring the cellular dynamics of differentiating hematopoietic cells. In 2022, Kwok et al. developed MQuad for the identification of informative mitochondrial variants from single-cell RNA (scRNAseq), DNA, or ATAC-seq data [14]. MQuad uses a binomial mixture model to identify mitochondrial genetic variants with both high sensitivity and specificity and is broadly applicable to various single-cell sequencing technologies complementing single-nucleotide and copy-number variation to extract finer clonal resolution. 

Here, we present MitoTrace, a user-friendly R package that enables the study of genetic variants and heteroplasmies in the mitochondrial genome in scRNAseq data. MitoTrace is based on the SAMtools framework [15], which was among the best performing variant calling methods for scRNAseq in a recent systematic comparison [16]. MitoTrace can be applied to both well-based and droplet-based scRNAseq data formats and thus provides universal compatibility with all major scRNAseq technologies. We demonstrate the validity and usage of our computational framework in a number of exemplary analyses of existing publicly available data sets. 

## 2. Materials and Methods

### 2.1. MitoTrace

MitoTrace takes as its input aligned BAM files as well as the mitochondrial genome sequence in FASTA format and makes good use of the Rsamtools R package which is based on the SAMtools framework [15] (Figure 1a). MitoTrace runs a pileup command which extracts the base at each genomic position within each individual read. The outputs are stored in two matrices. The first matrix contains the count of reads harboring the non-reference allele. The second matrix contains the read coverage at each genomic position for each sample (Figure 1b). The *MitoTrace()* function is used to calculate the read coverage and alternative allele counts across all positions in the mitochondrial genome. The *calc_allele_frequency()* function calculates the allele frequency of alternative alleles at each position in the mitochondrial genome. Those outputs can be used to perform standard analysis techniques, including heatmap visualization or dimension reduction via principal component analysis (Figure 1c). The *MitoDepth()* function is used to plot the read coverage across the mitochondrial genome (Figure 1d). 

The user needs to define the following parameters in MitoTrace’s main function. The user must provide the path to the mitochondrial reference sequence in FASTA format. This parameter extends MitoTrace to multiple species. For droplet-based scRNAseq data, there are many more unique barcodes than actual cells. Many barcodes represent empty droplets. In order to exclude these empty droplets, the user can either provide a list of barcodes or define a minimum detection cutoff to speed up processing. Please see our Github repository for detailed documentation and usage tutorials.

### 2.2. Preprocessing and Bulk DNAseq and scRNAseq Comparison

The prefetch and fastq-dump functions from the SRA Toolkit (version 2.9.6) were used to download the raw sequencing files from GEO. Raw RNA sequencing reads in FASTQ format were mapped to the human reference genome GRCh38 using STAR (version 2.7.0d) with the following parameters: --runThreadN 20 --runMode alignReads --outSAMtype BAM SortedByCoordinate --outFilterMultimapNmax 1 --readFilesCommand --genomeLoad LoadAndKeep --limitBAMsortRAM 200258540544. SAMtools (version 1.10) is used to extract the reads mapping to the mitochondrial genome. In order to identify uniquely discriminative variants, we performed the following analysis using the bulk DNAseq data. For each nucleotide position, we fitted a linear regression model as implemented in the R *lm()* function. The patient identifiers and alternative allele frequencies were defined as dependent and independent variables, respectively. Next, we selected variants with an alternative allele frequency greater than 0.9 in only one individual. The resulting *p*-values were used to sort variants displayed in Figure 2c.

### 2.3. Dimension Reduction and Random Forest Predictions

To perform dimension reduction on the mitochondrial variant profiles, we first identified the 500 most highly variable variant sites. Next, we calculated the Pearson correlation between all pairwise cell–cell combinations. One minus the absolute values of this correlation matrix was used as the input into the *Rtsne()* function of the R Rtsne package. “Perplexity” and “check_duplicates” parameters were set to 10 and FALSE, respectively. tSNE coordinates based on the transcriptome data were downloaded from the Appendix A [16]. We applied random forest models as implemented in the R package randomForest. A random forest classifier is an ensemble learning algorithm that combines multiple decision trees to classify data by taking a majority vote from the individual trees’ predictions [17]. The outcome variable and input features were the identifiers and alternative allele frequencies, respectively.

### 2.4. Identification of Informative Variants

To identify variants carrying information to classify the cells by patient, we used analysis of variance as implemented in the *aov()* R function. The patient identifiers and alternative allele frequencies were defined as dependent and independent variables, respectively. Variants with a *p*-value less than 1 × 10^−100^ were selected for display in Figure 3c.

### 2.5. Data Sets

The Chung et al. [18] data set profiled single cells from primary cancer and lymph nodes of the same patient bulk whole exon sequencing (WES) were downloaded from Gene Expression Omnibus (GEO) with the accession number GSE75688. Individual cells were captured using the Fluidigm C1 advanced microfluidic platform. This system provides automated procedures from total RNA isolation to preamplification of synthesized cDNAs. A total of 579 single-cell cDNAs were subjected to scRNAseq. The Darmanis et al. [19] data set profiled 3589 single brain tumor cells from 4 glioblastoma patients and was downloaded from the GEO with the accession number GSE84465. Brain tissue samples were collected from patients, then dissociated into single-cell suspensions. These single-cell suspensions were transferred to a fluorescence-activated cell sorting (FACS) buffer for single-cell sorting. After cDNA synthesis and library preparation, single cells were sequenced using paired-end sequencing mode.

The 10× Genomics scRNAseq data set was taken from the study by Kang et al. [20]. This data set contained an equal mixture of Jurkat and HEK293T cells and was downloaded from UCSF Box (https://ucsf.app.box.com/s/vg1bycvsjgyg63gkqsputprq5rxzjl6k) for the analysis in April 2023. The Enge et al. [21] data set profiled human pancreas cells from eight different human individuals, and 998 T cells were selected for the analysis and corresponding BAM files were downloaded from the GEO with the accession number GSE81547. Human islet samples were dissociated into single-cell suspensions. Those single cells were sorted on a FACS machine. Next, those sorted single cells were collected directly into 96-well plates for downstream single-cell RNA-seq libraries preparation.

## 3. Results

MitoTrace works on readily aligned single-cell and bulk RNA-seq data. The main MitoTrace function *MitoTrace()* takes as its input either a single BAM file or a list of BAM files and extracts the read coverage and alternative allele counts at every position of the mitochondrial genome. These data structures provide the foundation for the analyses of mitochondrial genetic variation outlined in this manuscript. 

### 3.1. Agreement between Bulk DNAseq and scRNAseq

To validate our method, we assessed the agreement between genetic variants detected by scRNAseq and bulk DNA sequencing (DNAseq) data. Chung et al. [18] studied tumor heterogeneity in eleven breast cancer patients and their data contained profiles from both bulk DNAseq and scRNAseq of the same individuals. After downloading the raw sequencing data, all reads were aligned to the human genome (GRCh38). Despite being sequenced at higher depth, the mitochondrial genome showed lower read density in the bulk DNAseq data compared to the scRNAseq data (Figure 2a).

Next, we applied MitoTrace to extract the read coverage and alternative allele count at each position. Stratification of alleles along these two axes, clearly separated germline mutations from potential sequencing noise or heteroplasmies in the bulk DNAseq data. Figure 2b illustrates this separation for one exemplary bulk DNAseq sample. Germline variants fall along the diagonal line (red points). In order to identify germline variants, we performed differential frequency analysis testing for mitochondrial variants with significantly different allele frequency across individuals (Appendix A). Indeed, analysis of these variants revealed common haplotype groups (Appendix A). We identified a total of 93 mitochondrial variants, which uniquely identified each patient in the bulk DNAseq data (Figure 2c). Using these informative variants, we set out to test if we could accurately predict the originating patient of each cell. Therefore, a random forest model was trained on these 93 alleles using MitoTrace-derived allele frequencies as its input to predict the individual patient on a cellular level. The model achieved 99% accuracy, demonstrating that the genetic information embedded in the bulk DNAseq data can be robustly recovered in the scRNAseq data using MitoTrace.

### 3.2. Discovery of Personal Genetic Variation

Next, we tested if MitoTrace can distinguish cells from different donors without first deriving informative variants from bulk DNAseq. Therefore, we re-analyzed publicly available data profiling the single-cell transcriptomes of four glioblastoma patients using Fluidigm C1 technology [19]. In the respective study, the authors provided a detailed dissection of glioblastoma cell types to improve our understanding of tumor formation and migration. We downloaded raw sequencing data and aligned all reads onto the human genome. Only the mitochondrial genome (average 6.2 reads per base pair) showed sufficient read coverage to enable variant calling on the cellular level compared to all nuclear chromosomes (average 4 × 10^−4^ reads per base pair) (Figure 3a). After applying MitoTrace to this data set, we calculated pairwise cell–cell distances based on the allele frequencies of the 500 most variable alleles. The visualization of this distance matrix in reduced dimensions using t-distributed stochastic neighbor embedding (tSNE) showed many subgroups and neighboring subgroups were derived from the same patient, suggesting that the mitochondrial allele frequencies can be used to distinguish patients (Figure 3b). Of note, the transcriptional profiles of the cells did not distinguish the patients and mainly clustered by cell type (Figure 3c,d). Therefore, the mitochondrial allele frequency profiles contained information that is complementary to the transcriptional information.

To identify personal variants without the integration of bulk DNA information, we statistically analyzed the allele frequencies. The analysis of variance identified 44 alleles with differential frequencies across the four patients (Figure 3e). These sites were characterized by the exclusive detection of the non-reference allele in practically all cells of a given patient, representing bona fide germline variants (Figure 3f, Appendix A). Indeed, analysis of these variants revealed common haplotype groups (Appendix A). Analogous to the analysis in Figure 1, using the frequencies at these alleles, a random forest model correctly predicted the patient with very high accuracy for every cell (99%). These results demonstrate that even without any bulk DNA sequencing information, scRNAseq can be used to detect informative alleles in an unsupervised fashion and subsequently assign individual cells to the patient of origin with very high accuracy.

### 3.3. Detection of Heteroplasmies

After having demonstrated that MitoTrace enables discovery of personal genetic variation, we set out to test whether MitoTrace can be used to discover mitochondrial heteroplasmies. Therefore, we reanalyzed publicly available scRNAseq data profiling human pancreas cells from eight different individuals [21]. In order to avoid bias introduced by comparison of different cell types, we restricted the analysis to one of the major cell groups in the data set consisting of 998 T cells. We downloaded the raw sequencing data and performed alignment. Next, we applied MitoTrace to the generated BAM files. Analogous to the analysis described above, we identified several personal variants. We then excluded these sites and focused on alleles with an average alternative allele frequency greater than 1% and less than 25% per individual. Among the remaining sites, we tested for differential allele frequencies across individuals. For example, we identified four sites that showed differential alternative allele frequency (Figure 4a). All four sites showed robust coverage, yet alternative allele frequency per cell was clearly distinct between different individuals (Figure 4b), demonstrating that MitoTrace can detect mitochondrial heteroplasmies specific to an individual.

### 3.4. Compatibility with Droplet-Based scRNAseq Data

In recent years, two types of scRNAseq technologies have been developed: well-based and droplet-based scRNAseq. All the studies analyzed in this manuscript so far used well-based technologies to generate the scRNAseq data. Well-based technologies are able to capture reads from the entire length of the transcripts but are limited by the number of wells on the chip [22]. The advent of droplet-based scRNAseq technologies enabled the molecular profiling of thousands of cells [23,24]. Droplet-based scRNAseq applies unique molecular identifiers (UMIs) to aggregate next-generation sequencing reads across molecules. These technologies capture molecules based on the polyA tail and therefore read coverage is limited to the 3′ end of a gene. 

The underlying data structure between well and droplet-based scRNAseq technologies is also different. In well-based scRNAseq data, the reads for each cell are stored in a separate BAM file. In droplet-based scRNAseq data, on the other hand, reads coming from all cells are stored in a single BAM file. Instead of generating one alignment file per cell, reads from all cells are saved in one alignment file with specific barcodes embedded in the file marking each cell. MitoTrace can be applied to both of these data formats and thus provides universal compatibility with all major scRNAseq technologies. 

To demonstrate that MitoTrace can be applied to droplet-based scRNAseq data, we downloaded a publicly available scRNAseq experiment provided by 10× Genomics. This platform captures RNA molecules using the polyA tail and therefore transcript read coverage is biased towards the 3′ end. Moreover, read coverage per cell is generally lower in droplet-based compared to well-based scRNAseq data. For these reasons, the power to call genetic variants is decreased in droplet-based compared to well-based scRNAseq technologies. 

Next, we set out to test whether MitoTrace could detect mitochondrial genetic variation in data derived from the 10× Genomics sequencing technology. Therefore, we analyzed an experiment which profiled an equal mixture of Jurkat and 293T cells. These two cell lines differ genetically and therefore can be used to test MitoTrace on droplet-based scRNAseq data [23]. After application of MitoTrace, we identified the 20 most highly variable alleles across all cells in an unsupervised manner (Figure 5a). Of note, some of these alleles showed large variation across all cells independent of cell line. For example, variant 12139T>A showed high variance in both Jurkat and 293T cells. However, the set of highly variable alleles also contained discriminatory sites, such as 3197T>C and 9698T>C, most likely representing germline differences between these two cell lines. Performing unsupervised clustering of these highly variable alleles separated the cells into two distinct clusters corresponding to Jurkat (blue) and 293T (red) cells (Figure 5b). These results demonstrate that MitoTrace robustly extracts genetic information from droplet-based scRNAseq data in an unsupervised fashion.

## 4. Discussion

Recent advancements in single-cell sequencing have opened up new possibilities for studying mitochondrial genetic variation at the single-cell level. In order to facilitate this analysis, MQuad [14], mGATK [13], and EMBLEM [11] have been developed. Both MQuad and EMBLEM require the use of specific variant calling tools such as GATK [25] or cellSNP [26] prior to analysis. Therefore, users cannot complete the analysis in a single programming language. In addition, mGATK is a command line tool and is disconnected from downstream analysis software. Finally, all these tools are implemented, to varying extents, in Python or Perl.

In this manuscript, we introduced MitoTrace, a novel method implemented using the widely adopted R statistical software. This distinctive feature offers users the convenience of performing the entire analysis within the same programming language, setting it apart from other existing tools. Our R-based tool integrates into a rich ecosystem of packages specifically designed for statistical analysis and data visualization including dimension reduction, random forest predictions, analysis of variance, and heatmap visualizations, as demonstrated in this manuscript.

Our study showcases the versatility of MitoTrace in detecting germline variations with remarkable accuracy, enabling the classification of cells by individuals. Furthermore, we demonstrated that MitoTrace is proficient in identifying mitochondrial heteroplasmies. It is worth noting that MitoTrace can readily be applied to various types of next-generation sequencing alignment data, including RNAseq, ATACseq, and DNAseq, both at the bulk and single-cell levels, making it a user-friendly tool for investigating mitochondrial genetics.

It is important to acknowledge that our analysis solely focuses on mitochondrial genetic variation, which might overlook informative genetic variation occurring in the nuclear genome. However, this limitation comes with notable benefits in terms of computational efficiency and speed. By narrowing the focus to the mitochondrial genome, MitoTrace streamlines the analysis process, making it more efficient for researchers. Given the high accuracies achieved in classifying cells originating from different individuals, we foresee the usage of MitoTrace as an efficient alternative for genetic barcoding with the purpose of demultiplexing cells from different donors or doublet identification in scRNAseq data. 

Estimating accurate heteroplasmy levels is challenging due to various factors such as sequencing errors, amplification biases, and the presence of nuclear mitochondrial DNA [27]. The following strategies can be considered to increase the accuracy of heteroplasmy identification and quantification. Firstly, researchers can generate technical replicates by performing independent sequencing runs of the same sample. Secondly, stringent quality control measures can be applied to remove low-quality reads and sequencing artifacts. Thirdly, sophisticated statistical models can be developed for heteroplasmy estimation by accounting for sequencing errors, base-calling quality scores, consensus sequence, and other relevant factors to provide more accurate estimates.

Our comprehensive tutorial available on GitHub (github.com/lkmklsmn/MitoTrace) provides various usage examples including the discovery of personal mitochondrial variants in SMART-seq2 data and the distinction of different cell lines in 10× Genomics data. In addition, we demonstrated that MitoTrace can reproduce the results from a previous study by Leif et al. [12] using just a few lines of R code.

In conclusion, MitoTrace is a user-friendly R package compatible with all scRNAseq technologies which accurately detects germline mitochondrial variation and mitochondrial heteroplasmies from bulk and scRNAseq data. Therefore, MitoTrace represents a powerful tool for researchers interested in studying mitochondrial genetics and thus presents a valuable contribution to the field of bioinformatics and mitochondrial biology.

## Figures and Tables

**Figure 1 genes-14-01222-f001:**
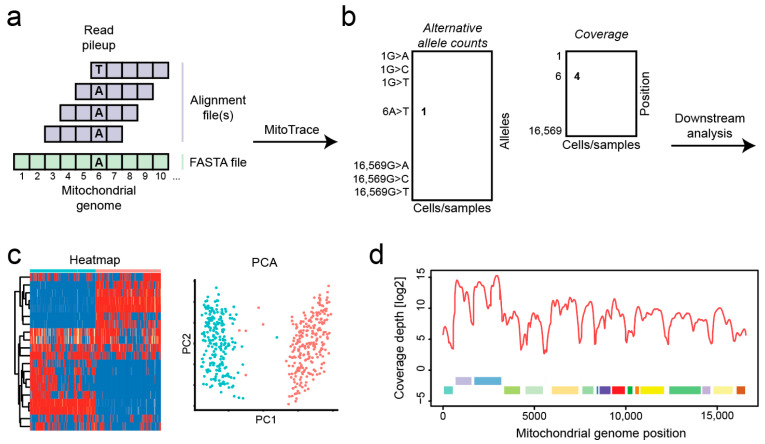
Overview of MitoTrace. (**a**) The sequencing reads are aligned to the mitochondrial reference genome. (**b**) MitoTrace generates pileup output for one or multiple alignment files. Each individual cell or sample file produces a separate pileup output, which is captured in the columns of the generated allele frequency matrix. (**c**) MitoTrace identifies mitochondrial genetic variation in scRNAseq sequencing data. The extracted allele frequencies can be visualized using heatmaps or subjected to dimension reduction such as principal component analysis. For example, MitoTrace can be used to differentiate cells from two distinct cell lines, Jurkat (blue) and HEK293T (red). (**d**) One built-in function plots the read coverage across the mitochondrial genome, including mitochondrial gene annotations.

**Figure 2 genes-14-01222-f002:**
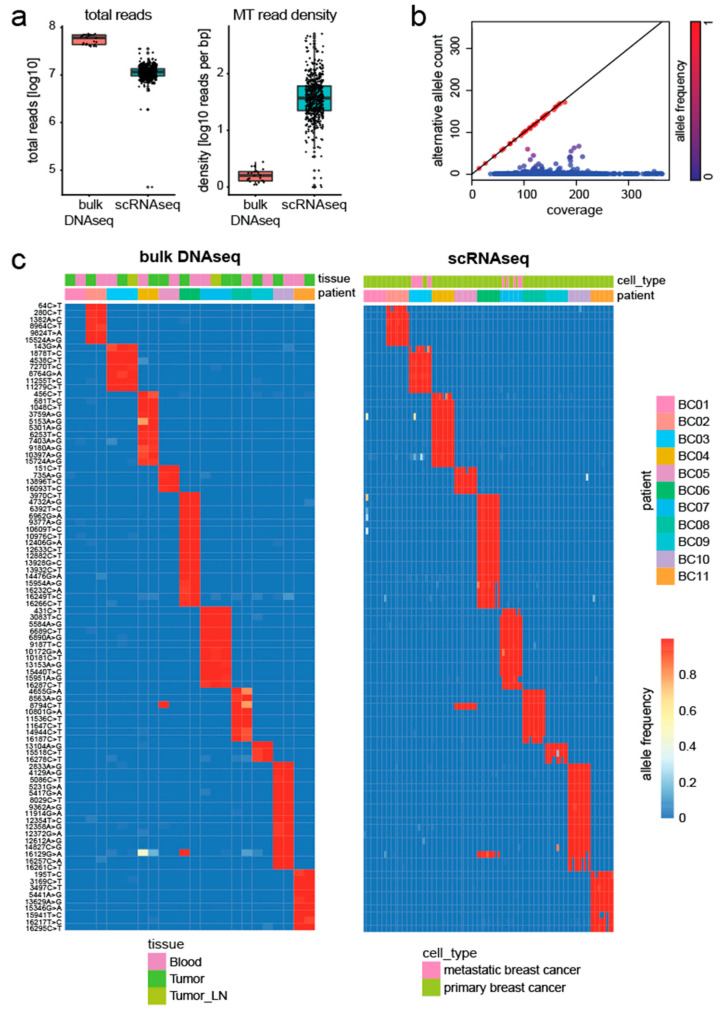
MitoTrace robustly recovers genetic information from scRNAseq data. (**a**) Boxplots display the number of total reads (left) and mitochondrial read density (right) for both bulk DNAseq and scRNAseq data. (**b**) Scatter plot stratifies mitochondrial alleles along the read coverage (*x*-axis) and alternative allele count (*y*-axis). Alleles are colored by allele frequency with red and blue colors reflecting high and low values, respectively. (**c**) Heatmaps display the allele frequency of bulk DNAseq samples (left) and scRNAseq (right). For both heatmaps, rows represent 93 discriminative alleles and are ordered by discriminatory power across the patients. Columns are ordered by patient. High and low allele frequencies are reflected by red and blue colors, respectively.

**Figure 3 genes-14-01222-f003:**
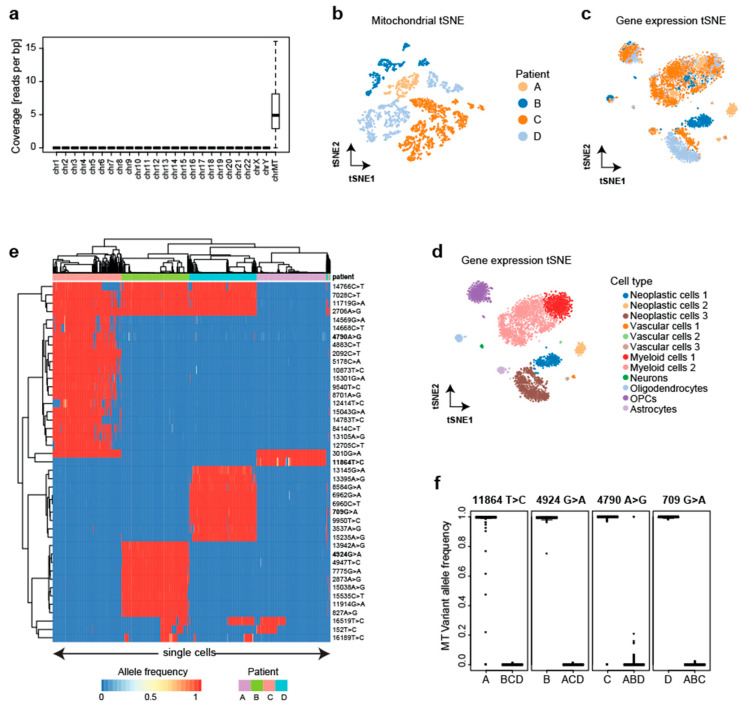
MitoTrace identifies personal genetic variants. (**a**) Boxplot shows the read coverage of individual cells across chromosomes measured in reads per base pair, highlighting high coverage on the mitochondrial chromosome. (**b**) tSNE visualization of the cell–cell distance matrix derived from mitochondrial allele frequency profiles readily separates the four patients. tSNE visualizations of the transcriptomic profiles colored by patient (**c**) and cell type identity (**d**) illustrate that cells cluster mainly by cell type. (**e**) Heatmap displays allele frequencies of discriminative variants across cells from the four patients. For visualization purposes, 100 cells were randomly sampled from each patient. Columns and rows represent individual cells and variant sites, respectively. Both columns and rows are ordered based on unsupervised hierarchical clustering. (**f**) Boxplots display the heteroplasmic profiles of four exemplary discriminative variants (highlighted in heatmap with bold font).

**Figure 4 genes-14-01222-f004:**
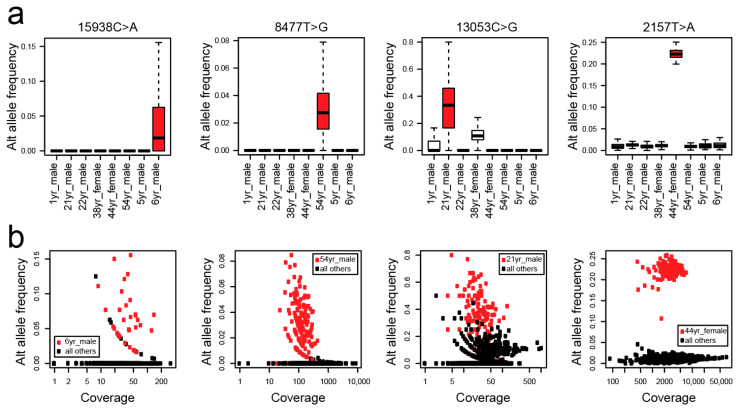
MitoTrace detects mitochondrial heteroplasmies. (**a**) Boxplots display alternative allele frequency (*y*-axis) for heteroplasmies 15938C>A, 8477T>G, 13053C>G, and 2157T>A (from left to right) of cells derived from various individuals (*x*-axis). Red color highlights individuals with robust heteroplasmies. (**b**) Scatterplots depict read coverage (*x*-axis) and alternative allele frequency (*y*-axis) of cells corresponding to the heteroplasmic sites in panel (**a**). Red color highlights cells derived from individuals 6yr_male, 54yr_male, 21yr_male, and 44yr_female matching the order in (**a**) (from left to right).

**Figure 5 genes-14-01222-f005:**
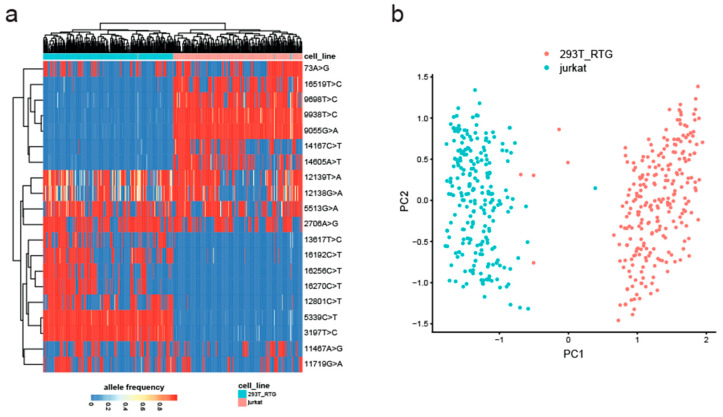
MitoTrace identifies mitochondrial genetic variation in 10× Genomics scRNAseq data. (**a**) Heatmap displays allele frequencies of the 20 most highly variable alleles across a mixture of Jurkat and 293T cells. Columns and rows represent individual cells and variant sites, respectively. Both columns and rows are ordered based on unsupervised hierarchical clustering. (**b**) Principal component analysis groups cells by origin. Jurkat and 293T cells are colored in blue and red, respectively.

## Data Availability

All the raw data used in this work can be accessed via our Github repository (github.com/lkmklsmn/MitoTrace).

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
