# Peer review of "MitoTrace: A Computational Framework for Analyzing Mitochondrial Variation in Single-Cell RNA Sequencing Data"

_genes, 2023, doi:10.3390/genes14061222_

Round 1

Reviewer 1 Report

In this manuscript Wang et al., reported MitoTrace as a computational resource to identify mitochondrial variation in scRNA seq data. This could be of an interest to the research community focusing not only single cell analysis but also mitochondrial domain. Following comments needs to be addressed,

1.   The introduction part focusing single cell RNAseq technology comparison does not seem necessary, only a line regarding this would be sufficient. Instead, the tools which are used to predict mitochondrial variation should be introduced in the introduction.

2.   Material and Methods section should focus on the strategy of MitoTrace, rather than the parameters it takes. The parameters should be moved to the supplement section or the tutorials section on GitHub.

3. Dataset details are not necessary in the materials sections. The links to datasets should be moved to supplement section.

4.       The detailed methodology or algorithm of MitoTrace should be provided in the methods section. It is better to present a flow diagram.

5.       The results section should include a part to compare the results with the existing tools in the field.  

Author Response

Reviewer 1:

In this manuscript Wang et al., reported MitoTrace as a computational resource to identify mitochondrial variation in scRNA seq data. This could be of an interest to the research community focusing not only single cell analysis but also mitochondrial domain. Following comments needs to be addressed,

Response:

We appreciate the reviewer’s support for the importance of our work. We believe that our MitoTrace computational framework is a novel and represents an important contribution to bioinformatics and mitochondrial biology domains. We have addressed all concerns point-by-point as described below and believe your comments have significantly improved the overall quality of our manuscript.

  1. The introduction part focusing single cell RNAseq technology comparison does not seem necessary, only a line regarding this would be sufficient. Instead, the tools which are used to predict mitochondrial variation should be introduced in the introduction.

Response:

We thank the reviewer for this suggestion. We reorganized the introduction and removed the discussion of the different single-cell RNAseq technologies. In the revised introduction, we now describe existing tools for the analysis of mitochondrial variation such as EMBLEM, MQuad, mgatk and summarize the function and application of those tools. The associated text has been added to the manuscript file, page 2.

  1. Material and Methods section should focus on the strategy of MitoTrace, rather than the parameters it takes. The parameters should be moved to the supplement section or the tutorials section on GitHub.

Response:

We thank the reviewer for the comments. We have removed the description of the parameters from the methods section, page 3.

  1. Dataset details are not necessary in the materials sections. The links to datasets should be moved to supplement section.

Response:

We thank the reviewer for the comments. We have moved the section describing the datasets used to the end of the Methods section, page 5.

  1. The detailed methodology or algorithm of MitoTrace should be provided in the methods section. It is better to present a flow diagram.

Response:

We thank the reviewer for this valuable suggestion. Following the reviewer’s suggestion, we have prepared a figure representing MitoTrace’s workflow (revised Figure 1). The new figure contains detailed information on the computational framework including read mapping, extracting allele frequencies and downstream analysis. The workflow is included in the revised manuscript as Figure 1. The figure legend has been added to the manuscript text, page 4.

  1. The results section should include a part to compare the results with the existing tools in the field.

Response:
We thank the reviewer for raising the comments. We have introduced other mitochondrial variation prediction tools such as EMBLEM, MQuad, mgatk and summarized the function and application of those tools in our introduction. However, MitoTrace is an R based tool that not only for statistical computing but also offers powerful data visualization capabilities. Besides, our R-based tool can be compatible with a wide range of comprehensive packages that have a strong active community. The comparison with existing tools has been added to the discussion section of the revised manuscript, page 11.

Reviewer 2 Report

In this manuscript by Wang et al., the authors introduce MitoTrace, a computational framework for the analysis of mitochondrial genetic variation in single-cell sequencing data. The authors showcase their tool across a variety of datasets and readily identify mitochondrial germline variants in single cells, which may be utilized to assign cells to specific donor groups. They further demonstrate how they identify likely somatic variants in single cells.

While largely sound, previous work and tools have already demonstrated similar capabilities, including Ludwig et al. (https://doi.org/10.1016/j.cell.2019.01.022) and Miller et al. (https://doi.org/10.1038/s41587-022-01210-8). This reviewer has the following comments.

Major comments

1. Considering similar and previously published tools, it is not clear to this reviewer how MitoTrace is bioinformatically distinct or compares to the tools mentioned above. Thus, the authors should elaborate/compare relevant differences and advantages.

2. The methods need to contain information about what exact methodology the analyzed data was generated with. For example, Smart-seq2 for well-based approaches and which 10x platform (3’ or 5’).

3. While the identification of germline mitochondrial variants appears trivial, how do the authors deal with variants that may present technical errors as discussed in Ludwig et al. (https://doi.org/10.1016/j.cell.2019.01.022) and Miller et al. (https://doi.org/10.1038/s41587-022-01210-8) which may lead to false-positive variant calls, in particular in scRNA-seq datasets. For example, Miller et al, the authors apply maegatk to use unique molecular identifiers (UMIs) to collapse multiple sequencing reads of the same starting transcript, creating a consensus call for every nucleotide based on the most common call and base quality. These would be important aspects to discuss and integrate into the manuscript.

4. Given other single-cell data modalities, including mitochondrial-single cell ATAC-seq https://doi.org/10.1038/s41587-020-0645-6, the authors should consider extending the application of MitoTrace to these data types.

Author Response

Reviewer 2:

In this manuscript by Wang et al., the authors introduce MitoTrace, a computational framework for the analysis of mitochondrial genetic variation in single-cell sequencing data. The authors showcase their tool across a variety of datasets and readily identify mitochondrial germline variants in single cells, which may be utilized to assign cells to specific donor groups. They further demonstrate how they identify likely somatic variants in single cells.

While largely sound, previous work and tools have already demonstrated similar capabilities, including Ludwig et al. (https://doi.org/10.1016/j.cell.2019.01.022) and Miller et al. (https://doi.org/10.1038/s41587-022-01210-8). This reviewer has the following comments.

We appreciate the reviewer’s support for the importance of our work. We believe that our MitoTrace computational framework is a novel and represents an important contribution to bioinformatics and mitochondrial biology domains. We have addressed all concerns point-by-point as described below and believe your comments have significantly improved the overall quality of our manuscript.

Major comments

  1. Considering similar and previously published tools, it is not clear to this reviewer how MitoTrace is bioinformatically distinct or compares to the tools mentioned above. Thus, the authors should elaborate/compare relevant differences and advantages.

Response:

We thank the reviewer for pointing this out. Following the reviewer’s suggestion, we introduce other mitochondrial variation analysis tools such as EMBLEM, MQuad, and mgatk in the introduction section of the revised manuscript, page 2. Given the 10 day deadline, we were unable to implement specific solutions to this. However, discussion of the comparison with existing tools has been added to the discussion section of the revised manuscript, page 12.

  1. The methods need to contain information about what exact methodology the analyzed data was generated with. For example, Smart-seq2 for well-based approaches and which 10x platform (3’ or 5’).

Response:

Following the reviewer’s suggestion we have added the requested information to the methods section of the revised manuscript.

  1. While the identification of germline mitochondrial variants appears trivial, how do the authors deal with variants that may present technical errors as discussed in Ludwig et al. (https://doi.org/10.1016/j.cell.2019.01.022) and Miller et al. (https://doi.org/10.1038/s41587-022-01210-8) which may lead to false-positive variant calls, in particular in scRNA-seq datasets. For example, Miller et al, the authors apply maegatk to use unique molecular identifiers (UMIs) to collapse multiple sequencing reads of the same starting transcript, creating a consensus call for every nucleotide based on the most common call and base quality. These would be important aspects to discuss and integrate into the manuscript.

Response:

We thank the reviewer for the suggestions and understand the important aspects of technical errors. MitoTrace focuses on extracting allele counts from the alignment files. Users have the flexibility to develop their own sophisticated statistical models to account for technical bias. Given the 10 day deadline, we were unable to implement specific solutions to this. However, discussion of technical errors has been added to the discussion section of the revised manuscript, page 12

  1. Given other single-cell data modalities, including mitochondrial-single cell ATAC-seq https://doi.org/10.1038/s41587-020-0645-6, the authors should consider extending the application of MitoTrace to these data types.

Response:

We thank the reviewer for the helpful suggestion. Indeed, MitoTrace is applicable to any alignment data, including other single-cell modalities such as ATAC-seq. We mention this feature in the discussion section of the revised manuscript, page 12.

Round 2

Reviewer 2 Report

The authors have revised their manuscript and while the methodology could be further developed (e.g., error correction) the current presentation is acceptable.